# Heterogeneous sulfate aerosol formation mechanisms during wintertime Chinese haze events: Air quality model assessment using observations of sulfate oxygen isotopes in Beijing

Jingyuan Shao[1,2], Qianjie Chen[2,3], Yuxuan Wang[4], Xiao Lu[1], Pengzhen He[5], Yele Sun[6], Viral Shah[2,7], Randall V. Martin[8], Sajeev Philip[9], Shaojie Song[7], Yue Zhao[10], Zhouqing Xie[5], Lin Zhang[1], and Becky Alexander[2]

[1]Laboratory for Climate and Ocean-Atmosphere Studies, Department of Atmospheric and Oceanic Sciences, School of Physics, Peking University, Beijing 100871, China

[2]Department of Atmospheric Sciences, University of Washington, Seattle, WA 98195, USA

[3]Department of Chemistry, University of Michigan, MI 48109, USA

[4]Department of Earth and Atmospheric Sciences, University of Houston, Houston, TX 77204, USA

[5]Anhui Province Key Laboratory of Polar Environment and Global Change, School of Earth and Space Sciences, University of Science and Technology of China, Hefei, Anhui 230026, China

[6]State Key Laboratory of Atmospheric Boundary Physics and Atmospheric Chemistry, Institute of Atmospheric Physics, Chinese Academy of Sciences, Beijing 100029, China

[7]School of Engineering and Applied Sciences, Harvard University, Cambridge, MA 02138, USA

[8]Department of Physics and Atmospheric Science, Dalhousie University, Halifax, Nova Scotia, Canada

[9]NASA postdoctoral program, NASA Ames Research Center, Moffett Field, CA, USA

[10]School of Environmental Science and Engineering, Shanghai Jiao Tong University, Shanghai 200240, China

*Correspondence to*: Becky Alexander (beckya@uw.edu), Lin Zhang (zhanglg@pku.edu.cn) and Zhouqing Xie (zqxie@ustc.edu.cn)

**Abstract**

Air quality models have not been able to reproduce the magnitude of the observed concentrations of fine particulate matter (PM$_{2.5}$) during wintertime Chinese haze events. The discrepancy has been at least partly attributed to low biases in modeled sulfate production rates due to the lack of heterogeneous sulfate production on aerosols in the models. In this study, we explicitly implement four heterogeneous sulfate formation mechanisms into a regional chemical transport model, in addition to gas-phase and in-cloud sulfate production. We compare the model results with observations of sulfate concentrations and oxygen isotopes ($\Delta^{17}O(SO_4^{2-})$) in the winter of 2014-2015, the latter of which is highly sensitive to the relative importance of different sulfate production mechanisms. Model results suggest that heterogeneous sulfate production on aerosols accounts for about 20% of sulfate production in clean and polluted conditions, partially reducing the modeled low bias in sulfate concentrations. Model sensitivity studies in comparison with the $\Delta^{17}O(SO_4^{2-})$ observations suggest that heterogeneous sulfate formation is dominated by transition metal ion catalyzed oxidation of SO$_2$.

## 1. Introduction

China has experienced rapid urbanization and industrialization in recent years, which has led to significant growth in concentration of PM$_{2.5}$ (particulate matter with aerodynamic diameter less than 2.5 μm) in Chinese megacities, particularly in Beijing (the capital of China) and surrounding areas (Zhang et al., 2016; Wang et al., 2014; Zhang et al., 2015). Extensive studies consistently show high PM$_{2.5}$ levels in winter due to increased coal combustion for heating and a stable atmospheric boundary layer (Sun, et al., 2016; Sun, et al., 2014; Liu et al., 2015a). The frequency and concentration of PM$_{2.5}$ pollution negatively impacts human health and atmospheric visibility and results in economic losses (Gao et al., 2015; Lelieveld, et al., 2015; Zhang et al., 2015). The Chinese government has implemented a series of policies to improve air quality, and as a result the annual average PM$_{2.5}$ concentration in Beijing decreased by ~20% from 2013 to 2017 (Sun et al., 2016; Zheng et al., 2018; Beijing Environment Protection Agency, 2018; Chinese State Council, 2013). Despite these improvements, PM$_{2.5}$ concentrations in Beijing still regularly exceed the Chinese National Ambient Air Quality Standard (CNAAQS, 35 μg m$^{-3}$ annual average) (Sun et al., 2016; Zhang et al., 2016; Beijing Environment Protection Agency, 2018).

Sulfate is one of the most important components of PM$_{2.5}$, representing 10-30% of PM$_{2.5}$ mass in eastern China (Huang et al., 2014; Zhang et al., 2013; Wang et al., 2014; Sun et al., 2016; Shi et al., 2017). Recent observations show that the sulfate mass fraction of PM$_{2.5}$ increases during haze pollution periods, indicating that sulfate is a key driver for severe haze events (Wang et al., 2014; Wang et al., 2016; Cheng et al., 2016; Li et al., 2017). Previous simulations have shown that most models fail to predict severe haze pollution in Beijing at least in part because of sulfate underestimation (Jiang et al., 2013; Park et al., 2014; Pozzer et al., 2012). Globally, sulfate production is dominated by the gas-phase oxidation of SO$_2$ by OH and aqueous-phase oxidation of S(IV) ($=SO_2 \cdot H_2O + HSO_3^- + SO_3^{2-}$) by H$_2$O$_2$, O$_3$, and O$_2$ catalyzed by transition metal ions

(TMI) in cloud droplets (Stockwell and calvert, 1983; Schwartz, 1987; Harris et al., 2013; Alexander et al., 2012; Chen et al., 2018). Heterogeneous sulfate production, which refers to aqueous-phase oxidation of S(IV) (=$SO_2 \cdot H_2O + HSO_3^- + SO_3^{2-}$) on the surface of and/or within the bulk pre-existing aerosols, is generally thought to be minor due to the low liquid water content of aerosols ($< 10^{-9}$ cm$^3$ cm$^{-3}$) compared to clouds ($10^{-8}$ to $10^{-6}$ cm$^3$ cm$^{-3}$) (Jacob, 2000). However, recent studies have shown that traditional gas- and aqueous-phase chemistry in cloud droplets cannot explain rapid sulfate production observed during haze, suggesting missing sulfate formation mechanisms on aerosols in the models (Zheng et al., 2015; Chen et al., 2016; Zhang et al., 2015; Wang et al., 2014; Cheng et al., 2016; Huang et al., 2014). These missing sulfate formation mechanisms on aerosols include heterogeneous oxidation of SO2 by NO2 (Cheng et al., 2016; Wang et al., 2016; Wang et al., 2018; Gao et al., 2016; Zhang et al., 2015) and O2 catalyzed by TMI (Li et al., 2017a; Li et al., 2011) and via a free radical chain mechanism (Huie and Neta, 1984; Hung and Hoffman, 2015). The importance of these heterogeneous reactions remains highly uncertain due in part to uncertainties regarding the aerosol liquid water content, pH, and ionic strength, all of which impact heterogeneous reaction rates (Herrmann et al., 2015; Cheng et al., 2016). In particular, ambient aerosol pH cannot be directly measured, and thus represents a large source of uncertainty (Hennigan et al., 2015).

Previous studies have calculated a large range of aerosol pH values (3.4 – 7.8) in Beijing using a thermodynamic model (ISORROPIA-II) (Fountoukis and Nenes, 2007). He et al. (2018) noted that the large differences in calculated aerosol pH depended on whether they assume aerosols exist in stable (Wang et al., 2016; He et al., 2018) or metastable state (Liu et al., 2017; Guo et al., 2017; He et al., 2018). Stable-state assumption allows for the simultaneous existence of solid and aqueous phases, while the metastable-state assumption allows for the existence of aqueous phase only by assuming salts to be supersaturated in aerosols (Fountoukis and Nenes, 2007). Aerosol pH values simulated assuming stable state are near neutral (pH of 7), much higher than when assuming metastable state (pH of 4-5) during haze events in China (Guo et al., 2017; Liu et al., 2017; He et al., 2018). However, a recent study calculated aerosol pH of around 4.6 during Chinese haze events for both the stable and metastable assumptions in ISORROPIA-II after fixing coding errors that impacted the stable state assumption (Song et al., 2018). This casts doubt on the existence of neutral aerosol pH during Chinese haze events and thus the importance of the NO2 oxidation pathway, which is not important under acidic conditions. Other factors, such as underestimate of NH3 emissions (Zhang et al., 2017) and Ca$^{2+}$ concentrations in aerosols (Shen et al., 2016) and the lack of consideration of organic acids in aerosol pH calculations (Wang et al., 2018), add additional uncertainties in estimates of aerosol pH and thus sulfate production rates and mechanisms in Chinese haze events.

The oxygen isotopic composition $\Delta^{17}O$ ($\approx \delta^{17}O - 0.52 \times \delta^{18}O$) of secondary sulfate ($\Delta^{17}O(SO_4^{2-})$) reflects the relative importance of different oxidation mechanisms in sulfate production because some of the oxidants transfer unique oxygen isotope signatures to the sulfate oxidation product (Savarino et al., 2000). Sulfate production via S(IV) oxidation by O3 and H2O2 leads to positive $\Delta^{17}O(SO_4^{2-})$ values of 9.8‰ and 0.7‰, respectively, while all other oxidants lead to $\Delta^{17}O(SO_4^{2-})$ at or near 0‰ (Table S1) (Vicars and Savarino, 2014; Savarino and Thiemens, 1999; Lee and Schwartz, 1983; Holt et al., 1981;

Dubey et al., 1997). Primary sulfate, both natural (dust and sea salt) and anthropogenic (coal and oil combustion), also has $\Delta^{17}O(SO_4^{2-})$ values equal to 0‰ (Dominguez et al., 2008; Lee et al., 2001). Once formed, sulfate in the atmosphere does not undergo further isotope exchange. Surface observations around the world show that $\Delta^{17}O(SO_4^{2-})$ ranges from 0 to 6‰ (Alexander et al., 2012; Alexander et al., 2005; Chen et al., 2016; Dominguez et al., 2008; Jenkins and Bao, 2006; Lee et al., 2001; Lee and Thiemens, 2001; McCabe et al., 2006; Patris et al., 2007; He et al., 2018). Due to the large positive enrichment of sulfate formed from $O_3$ oxidation ($\Delta^{17}O(SO_4^{2-})$ = 9.8 ‰) and the strong pH dependence of this aqueous-phase reaction, $\Delta^{17}O(SO_4^{2-})$ is highly sensitive to pH.

In this work, we implement four heterogeneous reactions for sulfate formation (via $H_2O_2$, $O_3$, $NO_2$, and TMI) into a global chemical transport model (GEOS-Chem) and compare the model results with observations of sulfate and $SO_2$ concentrations and $\Delta^{17}O(SO_4^{2-})$ in Beijing from October 2014 to January 2015. During this time period, Beijing held the Asia-Pacific Economic Cooperation (APEC) meeting from November 5-11, 2014. During and before APEC, $SO_2$ emissions in Beijing and its surrounding regions decrease due to strict emission controls to improve air quality (Zhang et al., 2016; Liu et al., 2015b). This paper is organized as follows. Section 2 describes the model simulations and the observations, and the method used to calculate heterogeneous sulfate production rates. Section 3 discusses model results with and without heterogeneous sulfate production considered in comparison with observed concentration and $\Delta^{17}O(SO_4^{2-})$. Section 4 discusses and explains the differences between our results and observations. Section 5 summarizes the main conclusions.

## 2 Methodology

### 2.1 GEOS-Chem model

We use the three-dimensional GEOS-Chem chemical transport model nested-grid version v10-01 (http://www.geos-chem.org/) to investigate sulfate formation mechanisms in Beijing, China between October 18, 2014 and January 17, 2015 (Wang et al., 2004; 2013; 2014; Zhang et al., 2015; 2016). The model has a horizontal resolution of 1/4° latitude by 5/16° longitude over East Asia (70°E - 140°E, 15°N - 55°N) and 47 vertical levels up to 0.01 hPa ($\approx$ 81 km). In the boundary layer where most heterogeneous sulfate production occurs, the vertical layer thickness is 120 - 150 m for the first 12 model layers (below 1700 m altitude). The model is driven by assimilated meteorological data from the NASA Goddard Earth Observing System (GEOS-FP), which has a temporal resolution of 3 hours (1 hour for surface quantities and mixing depths).

The model utilizes the global anthropogenic emission inventory EDGAR v4.2 (EC-JRC/PBL, http://edgar.jrc.ec.europa.eu/, 2011), overwritten by regional inventories such as the MIX Asian emission inventory over Asia (Li et al., 2017), EMEP over Europe (http://www.emep.int/index.html), and NEI2011 over the U.S. (https://www.epa.gov/ttn/chief/net/2011inventory.html). In particular, the MIX Asian emission inventory includes emissions of $SO_2$, $NO_x$, CO, $NH_3$ and NMVOC at a spatial resolution of 0.25°×0.25° for the year 2010 (Li et al., 2017; Geng et al.,

25   2017). Mineral dust aerosols are emitted in the model as described in Fairlie et al. (2007) and distributed in four size bins (radius of 0.1-1.0 μm, 1.0-1.8 μm, 1.8-3.0 μm, and 3.0-6.0 μm). In addition to natural dust, our model includes anthropogenic dust (radius of 0.1 - 1.0 μm) released from anthropogenic activities such as road-residential-commercial construction and combustion, following Phillip et al. (2017). Previous studies have suggested that anthropogenic dust accounts for about 25% of the $PM_{2.5}$ mass fraction in Beijing (Zhang et al., 2015; Phillip et al., 2017).

The sulfate-nitrate-ammonium aerosol system is fully coupled to oxidant chemistry (Park et al., 2004), with aerosol pH, ionic strength, and aerosol water content (AWC) calculated from the ISORROPIA-II thermodynamic equilibrium model (Fountoukis and Nenes, 2007) that was implemented into GEOS-Chem by Pye et al. (2009). In the standard model (Run_Std), sulfate is produced from gas-phase oxidation of $SO_2$ by OH, aqueous-phase oxidation of S(IV) ($=SO_2 \cdot H_2O + HSO_3^- + SO_3^{2-}$)

35   by $H_2O_2$ and $O_3$ in cloud droplets, and heterogeneous oxidation on sea-salt aerosols by $O_3$ (Alexander et al., 2005). Primary anthropogenic emissions of sulfate constitute 3.1% of total anthropogenic sulfur emissions in China and 1.5% - 3.5% elsewhere. Sulfate is removed from the atmosphere via dry (Zhang et al., 2001) and wet (Liu et al., 2001) deposition, with a global lifetime of about 4 days (Alexander et al., 2005).

40   We performed three simulations at high horizontal resolution (1/4°×5/16°) and seven sensitivity simulations at low horizontal resolution (4°×5°) to investigate sulfate formation mechanisms in Beijing, as summarized in Table 1. In the model simulation Run_TMI, we implemented the in-cloud TMI-catalyzed aqueous-phase S(IV) oxidation by $O_2$ chemistry into the model, which is thought to be one of the most important sulfate formation pathways during the North Hemisphere winter (Huang et al., 2014; Harris et al., 2013; Alexander et al., 2009; Sofen et al., 2011). The parameterization of TMI-catalyzed S(IV)

45   oxidation in cloud for GEOS-Chem follows Alexander et al. (2009), but with reduced solubility of trace metals Fe and Mn derived from natural dust (from 1% to 0.45% for Fe and from 50% to 5% for Mn) to better match observations (Desboeufs et al., 2001; 2005; Chuang et al., 2005). In the model, Fe from natural dust ($[Fe]_{nat}$) is 3.5% of total dust mass and Mn from natural dust ($[Mn]_{nat}$) is a factor of 50 lower than $[Fe]_{nat}$ (Alexander et al., 2009). Anthropogenic Mn ($[Mn]_{ant}$) and Fe ($[Fe]_{ant}$) are scaled to the abundance of primary anthropogenic sulfate due to common sources and atmospheric lifetimes. $[Mn]_{ant}$ is

50   1/300 of primary sulfate concentration and $[Fe]_{ant}$ is 10 times that of $[Mn]_{ant}$ as described in Alexander et al. (2009). Only soluble Fe and Mn in the oxidation states Fe(III) and Mn(II) catalyze S(IV) oxidation. For Fe, we assume solubility of 10% for $[Fe]_{ant}$ and 0.45% of $[Fe]_{nat}$ in cloud water, respectively, with 10% in the form of Fe(III) during daytime and 90% at night. For Mn, we assume a solubility of 50% for $[Mn]_{ant}$ and 5% for $[Mn]_{nat}$ in cloud water, respectively, with 100% in the form of Mn(II). After modification, the average aqueous-phase concentration of Fe(III) in cloud water during our studying period

55   is 2.9 μM and Mn(II) is 1.3 μM in the model, which is consistent with previous work (Fe(III): 0.6-7.4 μM; Mn(II): 0.4-1.7 μM) (He et al., 2018; Shen et al., 2012; Guo et al., 2012) (see Text S3 in the Supplement for more details).

In the model run Run_Het, we added four heterogeneous sulfate production mechanisms (via $H_2O_2$, $O_3$, $NO_2$, and TMI) on

aerosols into the model, in addition to TMI-catalyzed oxidation in clouds. Implementation of these reactions in the model is described in Sect. 2.2. In Run_Het, heterogeneous sulfate production on aerosols only occurs when relative humidity (RH) is greater than 50%, effectively assuming that aerosol water content is too low for sufficient heterogeneous sulfate production at RH < 50%. When RH < 50%, aerosols are assumed to remain crystallized until reaching the deliquescence relative humidity (DRH). This is consistent with observations in previous studies showing that sulfate production rates in Chinese haze are positively correlated with RH (Sun et al., 2013; Zhang et al., 2015; Wang et al., 2016). $Ca^{2+}$ and $Mg^{2+}$ cations from dust (both natural and anthropogenic) are included in the aerosol thermodynamic calculations (aerosol pH, aerosol water content, ionic strength). We assume that $Ca^{2+}$ and $Mg^{2+}$ cations constitute 3.0% and 0.6% of the dust by mass, respectively, based on observations near East Asian dust source regions (Fairlie et al., 2010). In addition, we considered the impacts of acidity and ionic strength on TMI-catalyzed reaction rates following Cheng et al. (2016) (Table 2), since the ionic strength of aerosol liquid water can reach 20 M during polluted periods (He et al., 2018; Herrmann et al., 2015). We performed seven sensitivity studies based on Run_Het but with prescribed values of aerosol pH to examine the dependence of model results on aerosol pH alone (Table 1).

For all model simulations, sulfate produced from each oxidation pathway is labeled as a separate "tracer" in the model with a corresponding $\Delta^{17}O(SO_4^{2-})$ values (9 sulfate tracers in total, Table S1) as originally described in Alexander et al. (2005). Primary anthropogenic sulfate is also included as a separate tracer in the model with $\Delta^{17}O = 0‰$ (Lee et al., 2002). The details for calculating $\Delta^{17}O(SO_4^{2-})$ in the model are described in the supplement (Text S1).

## 2.2 Heterogeneous sulfate production on aerosols

Heterogeneous S(IV) oxidation in the model occurs on all aerosol types, including sulfate-nitrate-ammonium, dust, black carbon, organic carbon, and sea-salt aerosols. The heterogeneous sulfate production rate on aerosol ($P_{het}$) is calculated in the model assuming a first-order loss of $SO_2$ or oxidant (depending on which is the rate limiting step) via uptake by the aerosol (Eq. 1).

$$P_{het} = k[SO_2 \text{ or oxidant}] \qquad (1)$$

The first-order loss rate constant ($k$, s$^{-1}$) is calculated using the reaction probability formulation in Jacob (2000) (Eq. 2).

$$k = \left(\frac{r_a}{D_g} + \frac{4}{v\gamma}\right)^{-1} A \qquad (2)$$

where $r_a$ is the radius of the specific type of aerosol (cm); $A$ is the total aerosol surface area per unit volume of air for the specific type of aerosol (cm$^2$ cm$^{-3}$); $v$ is the mean molecular speed of $SO_2$ or the oxidant (cm s$^{-1}$); $D_g$ represents the gas-phase molecular diffusion coefficient of $SO_2$ or the oxidant (cm$^2$ s$^{-1}$) calculated as:

$$D_g = \frac{9.45 \, l \, 10^{17} \times \sqrt{T \times (3.47 \times 10^{-2} + (1/M))}}{\rho_{\text{air}}} \quad (3)$$

where $T$ is air temperature (K), $\rho_{\text{air}}$ is air density (molecule m$^{-3}$), and $M$ represents the molar mass of SO$_2$ or the oxidant (g mol$^{-1}$). The reaction probability ($\gamma$) is defined as the probability that a molecule impacting the aerosol surface undergoes a chemical reaction (Ravishankara, 1997; Jacob, 2000). Due to limited understanding of sulfate formation on aerosols, chemical transport models typically calculate heterogeneous sulfate production rate on aerosols by assuming the bulk first-order uptake of SO$_2$ and using a wide range of $\gamma$ values ($10^{-4}$ - $10^{-1}$) (Wang et al., 2014; Zhang et al., 2014). However, the relative contribution of different sulfate production mechanisms, which is important to inform air pollution mitigation efforts, cannot be determined with this simplified approach.

In this study, we use a more specific approach to calculate $\gamma$ for each heterogeneous sulfate production mechanisms following Jacob (2000) and Ammann et al. (2013) (Eq. 4).

$$\gamma = \left[ \frac{1}{\alpha} + \frac{v}{4K^* R T \sqrt{D_a k_{chem}}} \cdot \frac{1}{f_r} \right]^{-1} \quad (4)$$

where $\alpha$ is the mass accommodation coefficient (unitless); $k_{chem}$ is the pseudo first-order aqueous-phase chemical rate constant between S(IV) and the oxidant (O$_3$, H$_2$O$_2$, NO$_2$ or O$_2$) (s$^{-1}$) (Table 2); $D_a$ is the aqueous-phase molecular diffusion coefficient of SO$_2$ or the oxidant (cm$^2$ s$^{-1}$); $K^*$ is the effective Henry's law constant of SO$_2$ or the oxidant (M atm$^{-1}$); $R$ is the universal gas constant (L atm mol$^{-1}$ K$^{-1}$), and $f_r$ is reacto-diffusive correction term that compares the radius of aerosols ($r_a$) with the reacto-diffusive length scale of the reaction ($l$):

$$f_r = \coth\frac{r_a}{l} - \frac{l}{r_a} \quad (5)$$

$$l = \sqrt{\frac{D_a}{k_{chem}}} \quad (6)$$

In the model, the heterogeneous sulfate production rate from the TMI-catalyzed reaction is calculated as first-order uptake in SO$_2$. All other heterogeneous sulfate production pathways are calculated as first-order uptake in the oxidant (H$_2$O$_2$, O$_3$, and NO$_2$). This is based on whether the heterogeneous sulfate production on aerosol is limited by the availability of SO$_2$ or the oxidant. For aerosol pH values less than 6, heterogeneous sulfate production rates calculated as a first-order loss in SO$_2$ or the oxidant are similar (Figure S1(b)). For aerosol pH values greater than 6, heterogeneous sulfate production rates calculated as a first-order loss in SO$_2$ are higher than those calculated as a first-order loss in the oxidants O$_3$ and NO$_2$. The reaction rate for S(IV) oxidation by O$_3$ and NO$_2$ increases with increasing pH, and at high pH values $\gamma$ is limited by the mass accommodation coefficient and becomes independent of pH. The mass accommodation coefficients for O$_3$ ($2\times10^{-3}$) and NO$_2$

$(2 \times 10^{-4})$ are much lower than for $SO_2$ (0.23). The mass accommodation coefficient for $H_2O_2$ (0.11) is similar to $SO_2$, and $\gamma$ for the reaction of S(IV) with $H_2O_2$ was limited by the oxidant concentration. More details on first-order loss rates are described in Text S2.

In addition to the model simulations described in Table 1, we have also examined heterogeneous oxidation of $SO_2$ by $O_2$ on the surface of acidic microdroplets (Hung and Hoffman, 2015) and by HOBr (Chen et al., 2016; 2017). The results are described in the supplementary material (Text S2).

### 2.3 Observations of sulfate concentrations and oxygen isotopic composition

Between 17 October 2014 and 20 January 2015, PM$_{2.5}$ samples were collected every 12 hours for daytime (08:00-20:00 Beijing Time) and nighttime (20:00-08:00 Beijing Time) conditions at the campus of University of the Chinese Academy of Sciences (40.41°N, 116.68°E, 20 m from the ground) in Beijing, around 60 km northeast of downtown. The oxygen-17 excess of sulfate ($\Delta^{17}O(SO_4^{2-})$) of these PM$_{2.5}$ samples were measured at University of Washington, Seattle. A detailed description of sampling and measurements of $\Delta^{17}O(SO_4^{2-})$ can be found in He et al. (2018). Observations of hourly sulfate concentrations were conducted at the Institute of Atmospheric Physics (IAP), Chinese Academy of Sciences (39.99°N, 116.37°E) at ground level (~4 m), an urban site in the north of Beijing, as described by Sun et al. (2016). Sulfate concentrations in submicron aerosols (PM$_1$) were measured by an Aerodyne High Resolution Aerosol Mass Spectrometer (HR-AMS). The difference of $SO_4^{2-}$ concentration between PM$_{2.5}$ and PM$_1$ is small because most sulfate exists in fine aerosols (Guo et al., 2014). A comparison of $SO_4^{2-}$ concentration in PM$_{2.5}$ and PM$_1$ during 22 November – 29 November 2017 is shown in the supplement (Figure S2). Surface PM$_{2.5}$, $SO_2$, $NO_2$, and $O_3$ measurements (Figures 1 and S6) are from the China National Environmental Monitoring Center (http://106.37.208.233:20035/) with 12 sites in Beijing, including 8 urban and 4 suburban sites.

## 3 Results

### 3.1 Observed PM$_{2.5}$ and sulfate concentrations and $\Delta^{17}O(SO_4^{2-})$

Figure 1 shows the time series of observed concentrations of PM$_{2.5}$ and sulfate from 17 October 2014 to 20 January 2015, along with temperature and relative humidity in Beijing from GEOS-FP. A previous study showed that the GEOS-FP meteorological data for temperature and relative humidity are in good agreement with the ground-based measurements in Beijing ($R^2 > 0.9$) (Zhang et al., 2016). The CNAAQS defines the 24h average air quality levels as excellent (PM$_{2.5}$ = 0-35 $\mu$g m$^{-3}$), good (PM$_{2.5}$ = 35-75 $\mu$g m$^{-3}$), light (PM$_{2.5}$ = 75-115 $\mu$g m$^{-3}$), moderate (PM$_{2.5}$ = 115-150 $\mu$g m$^{-3}$), heavy (PM$_{2.5}$ = 150-250 $\mu$g m$^{-3}$), and severe (PM$_{2.5}$ > 250 $\mu$g m$^{-3}$). Using this metric, 10 time periods are categorized as heavy or severe (>150 $\mu$g m$^{-3}$) during 17 October 2014 – 20 January 2015, and we refer to these as "Heavy Polluted Period" (HPP). Another 10 time periods were in the excellent to good category and we refer to these days as "Clean Period" (CP). The average relative humidity (RH) from GEOS-FP during HPP was 60±11%, much higher than the average RH during CP (42±10%). Higher

RH can accelerate the rates of conversion of $SO_2$ and $NO_2$ to $SO_4^{2-}$ and $NO_3^-$, respectively, contributing to increases in $PM_{2.5}$ concentrations (Wang et al., 2016; Hua et al., 2015).

The observed $SO_4^{2-}$ concentrations show a similar variation as $PM_{2.5}$, increasing from 2.1±1.8 µg m$^{-3}$ in CP to 25.9±11.3 µg m$^{-3}$ in HPP with a mean of 11.5±7.3 µg m$^{-3}$ during the entire measurement period. The mass fraction of $SO_4^{2-}$ to $PM_{2.5}$ ranged from 5% to 19%, varying from a mean of 8±2% in CP to 13±2% in HPP. Observed sulfate concentrations shown in Figure 1 are 15% lower on average during HPP than those reported in He et al. (2018) because the $SO_4^{2-}$ concentrations shown here represent $PM_1$ instead of $PM_{2.5}$, which is in good agreement with previous studies (Sun et al., 2013; 2014; 2016) (Fig. S2). Figure 1 also shows the observed sulfur oxidation ratio (SOR), defined as the molar ratio of sulfate over the sum of sulfate and $SO_2$ ($SO_4^{2-}/(SO_2 + SO_4^{2-})$) (Colbeck and Harrison, 1984). The observed SOR increases from CP (9±6%) to HPP (32±18%), consistent with increased sulfate production rates during HPP.

Figures 2 and S4 show the $\Delta^{17}O(SO_4^{2-})$ observations, 24 measurements represent HPP and 10 represent CP. The $\Delta^{17}O(SO_4^{2-})$ values are similar in HPP and CP, 0.9±0.1‰ and 0.9±0.5‰, respectively. A more detailed description of the $\Delta^{17}O(SO_4^{2-})$ observations can be found in He et al. (2018).

### 3.2 Analyses of sulfate formation pathways

### 3.2.1 Sulfate formation in the standard model

Figure 1 compares the measured and modeled $PM_{2.5}$ and sulfate concentration in Beijing. The standard model (Run_Std) generally captures the temporal variations of $PM_{2.5}$ and $SO_4^{2-}$ observations during the entire sampling period, but underestimates the $PM_{2.5}$ and $SO_4^{2-}$ observations during HPP by 28% and 64%, respectively. Modeled sulfate concentrations increase from 1.5±1.2 µg m$^{-3}$ in CP to 8.9±3.1 µg m-3 in HPP, which is a much smaller enhancement compared to observations (from 2.1±1.8 µg m$^{-3}$ in CP to 25.9±11.3 µg m$^{-3}$ in HPP). The model simulated sulfate mass fractions in $PM_{2.5}$ are 8±2% in CP and 7±2% in HPP. The model fails to reproduce increases in observed sulfate mass fraction from CP (8±2%) to HPP (13±2%). This is consistent with previous modeling studies, suggesting missing sulfate formation pathways (Wang et al., 2014; Zhang et al., 2015). The model also underestimates the sulfur oxidation ratio (SOR) observations during HPP by 53%. This further suggests that the modeled $SO_2$ oxidation rate is too slow.

Figure 2 compares observed and modeled $\Delta^{17}O(SO_4^{2-})$. The modeled hourly $\Delta^{17}O (SO_4^{2-})$ values were averaged at 12h intervals for comparison with the observations. The simulated $\Delta^{17}O(SO_4^{2-})$ values in Run_Std range from 0.02‰ to 1.5‰ with a mean of 0.5±0.1‰. Unlike the observations, modeled $\Delta^{17}O (SO_4^{2-})$ values in Run_Std during HPP (0.4±0.1‰) are lower than those during CP (0.6±0.1‰) due to higher fractional contribution of in-cloud $H_2O_2$ and $O_3$ oxidation pathways during CP, as discussed below. Run_Std underestimates the $\Delta^{17}O(SO_4^{2-})$ observations (by 44% on average), particularly

during HPP (by 53% on average).

Figure 3 shows modeled spatial distribution of sulfate concentrations over China, and the fractional contribution of each sulfate formation pathway to total sulfate abundance in Beijing. The average simulated sulfate concentration in Beijing in Run_Std is around 6.2 $\mu g\,m^{-3}$ (Figure 3a), smaller than the observations (11.5 $\mu g\,m^{-3}$) over the study period. Figure 3 also shows the model calculated average contribution of each sulfate production pathway to total sulfate concentration at the surface in Beijing during HPP and CP. In Run_Std, sulfate produced by gas-phase oxidation of $SO_2$ by OH dominates the total sulfate abundance (59.7%, 5.3 $\mu g\,m^{-3}$) in HPP. Primary sulfate represents the second most important contributor (33.6%, 3.0 $\mu g\,m^{-3}$) despite the fact that primary sulfate is only 3% of total anthropogenic sulfur emissions. The high fraction of primary anthropogenic sulfate reflects the relatively slow oxidation rate of $SO_2$ in the model. In-cloud sulfate production contributes only 7% of total sulfate abundance at the surface during HPP. For clean days, primary sulfate dominates surface sulfate concentrations (49.4%, 0.7 $\mu g\,m^{-3}$), suggesting an even slower $SO_2$ oxidation rate in the model during CP compared to HPP. Gas-phase production via OH oxidation is the second most important contributor (26.1%, 0.4 $\mu g\,m^{-3}$) during CP. Gas-phase oxidation of $SO_2$ by OH is more important in HPP than in CP because modeled OH concentrations during HPP are much higher than in CP, consistent with observations of high OH during polluted wintertime conditions in Beijing (Tan et al., 2018). Higher modeled OH in HPP compared to CP is due to higher nitrous acid (HONO) levels during HPP (Figure S3) resulting from heterogeneous uptake of $NO_2$ to produce $HNO_3$ and HONO in the model. This is consistent with observation-based studies in Beijing showing that OH production from HONO photolysis is 10 times higher than that from $O_3$ photolysis in winter (Hendrick et al., 2014) and that $NO_2(g)$ dissolution in acidic aerosol water is a source of HONO (Li et al., 2018). In-cloud sulfate production contributes 24.5% of total sulfate abundance during CP, much higher than HPP (6.9%) due to higher modeled $H_2O_2$ and $O_3$ in CP.

### 3.2.2 Transition metal ion catalyzed oxidation of S(IV) in clouds

The in-cloud concentration of soluble $Fe^{3+}$ and $Mn^{2+}$ determines the rate of sulfate formation via the TMI-catalyzed oxidation pathway, but large uncertainties exist in estimates of soluble $Fe^{3+}$ and $Mn^{2+}$ due to lack of observations. Adding aqueous-phase TMI-catalyzed S(IV) oxidation by $O_2$ in cloud droplets in Run_TMI increases the average sulfate concentration in Beijing during the entire measurement period from 6.2 $\mu g\,m^{-3}$ to 8.3 $\mu g\,m^{-3}$ due to increases in the in-cloud sulfate production rate. However, the model still underestimates observations of PM$_{2.5}$ (-35%), sulfate (-48%), and SOR (-40%) during HPP. Sulfate from TMI-catalyzed oxidation dominates in-cloud sulfate production and accounts for up to 28.3% of total sulfate abundance during HPP, but only 8.1% during CP in Beijing (Figure 3b). The lower contribution of TMI-catalysis during CP is due to lower concentrations of Fe and Mn in the model during CP. The largest enhancement in sulfate abundance after adding the in-cloud TMI pathway occurs in Sichuan basin (around 6.5 $\mu g\,m^{-3}$), where simulated anthropogenic Fe and Mn from coal fly ash (Figure S4) and $SO_2$ are high due to high $SO_2$ emissions (Zhang et al., 2009) combined with stagnant air

and high relative humidity all year (Huang et al., 2014). After adding the in-cloud TMI oxidation pathway, the average modeled $\Delta^{17}O(SO_4^{2-})$ decreased from 0.5‰ to 0.4‰ in Beijing because the TMI oxidation pathway leads to $\Delta^{17}O(SO_4^{2-}) = 0‰$ (Figure 2), which makes the discrepancy between modeled and observed $\Delta^{17}O(SO_4^{2-})$ (0.9 ‰) even larger.

### 3.2.3 Heterogeneous sulfate formation on aerosols

Adding four heterogeneous S(IV) oxidation mechanisms by $H_2O_2$, $O_3$, $NO_2$ and TMI-catalyzed $O_2$ on aerosols in Run_Het increases the average $SO_4^{2-}$ concentrations in Beijing (from 8.3 μg m$^{-3}$ in Run_Std to 9.8 μg m$^{-3}$ in Run_Het) (Figure 1c). Modeled heterogeneous sulfate production represents 21.6% of total surface sulfate concentrations in HPP and 19.8% in CP (Figure 3c). Modeled daily-mean aerosol pH ranged from 3.0 to 5.4 with a mean of 4.3 over the entire time period in Beijing (Figure 4), which is consistent with recent estimates (pH=4.2-4.7) (Guo et al, 2017; Song et al, 2018; Liu et al., 2017). Heterogeneous sulfate production on aerosols is dominated by TMI-catalyzed $O_2$ oxidation in both HPP (69%) and CP (67%) (Figure 3c). S(IV) oxidation by $O_3$ is the second most important heterogeneous oxidation pathway, accounting for 19% of total heterogeneous sulfate formation in both HPP and CP. S(IV) oxidation by $H_2O_2$ (6% in both HPP and CP) and $NO_2$ (6% in HPP and 8% in CP) represent a minor heterogeneous sulfate production pathway. Previous studies suggested that oxidation of $SO_2$ by $NO_2$ in aerosol water dominates heterogeneous sulfate formation in Beijing during winter (Wang et al., 2016; Cheng et al., 2016) at neutral aerosol pH. However, subsequent studies showed that these high aerosol pH values were unlikely during wintertime Beijing haze events and they calculated aerosol pH values in the range of 4.2-4.7 (Guo et al, 2017; Song et al, 2018; Liu et al., 2017).

After adding the heterogeneous S(IV) oxidation pathways, the average modeled $\Delta^{17}O(SO_4^{2-})$ increased from 0.5±0.5‰ to 0.8±0.7‰ in Beijing due to the increased importance of S(IV) oxidation by $O_3$. Although the average modeled $\Delta^{17}O(SO_4^{2-})$ in Run_Het is similar to the observations (0.9±0.3‰), the modeled median (0.5‰) largely underestimates the observed median (1.0‰) (Figure 2), and the majority of the modeled data underestimates the observed $\Delta^{17}O(SO_4^{2-})$ (Figure S5).

Compared to observations in Beijing, mean model biases decrease from -28% to -26% for PM$_{2.5}$, -45% to -21% for sulfate concentration, from -29% to -11% for SOR, and from -45% to -15% for $\Delta^{17}O(SO_4^{2-})$ for the entire measurement period for Run_Het relative to Run_Std. Model biases during HPP decrease from -38% to -32% for PM$_{2.5}$, -65% to -40% for sulfate concentration, -53% to -28% for SOR, and from -50% to -5% for $\Delta^{17}O(SO_4^{2-})$. The largest sulfate enhancements due to heterogeneous sulfate formation occur in megacities in eastern China and Sichuan basin (Figure 3), where both $SO_2$ and aerosols abundances are highest. In addition, high anthropogenic emissions of $Fe^{3+}$ and $Mg^{2+}$ favor sulfate production catalyzed by TMIs. In Run_Het, the fractional contribution of each sulfate production mechanism in Beijing during HPP is 34% for gas-phase oxidation of $SO_2$ by OH, 22% for TMI-catalysis in cloud, 21% for the four heterogeneous oxidation pathways, and 20% for primary sulfate, respectively (Figure 3c). The remaining 3% of sulfate is formed via in-cloud sulfate

production from $H_2O_2$ and $O_3$. For clean days (CP), primary sulfate still dominates surface sulfate concentrations (39%) in Run_Het, with gas-phase production via OH oxidation the second most important contributor (22%) and the added heterogeneous sulfate formation pathways accounting for 22%.

## 4. Discussion

The model results demonstrate that implementation of heterogeneous sulfate formation pathways on aerosols reduces modeled low biases in both concentration and oxygen isotopic signature of sulfate ($\Delta^{17}O(SO_4^{2-})$), and suggests that TMI-catalysis dominates heterogeneous sulfate production. However, the model is still biased low in both metrics, indicating that the model is still underestimating sulfate production rates. The modeled underestimate in $\Delta^{17}O(SO_4^{2-})$ reveals an underestimate in the $O_3$ oxidation pathway. Since the relative contribution of heterogeneous sulfate formation pathways are sensitive to aerosol pH, we investigated the influence of aerosol pH using a series of sensitivity simulations with prescribed aerosol pH values between 2 and 8 (Table 1). Figure 5 shows modeled sulfate concentrations produced from different oxidation pathways and resulting $\Delta^{17}O(SO_4^{2-})$ in Beijing averaged over HPP assuming different aerosol pH values. Heterogeneous sulfate formation represents over 50% of total sulfate formation when pH > 5 and when pH < 4. At aerosol pH < 4, heterogeneous sulfate formation is dominated by the TMI pathway because of the high solubility of Fe and Mn at low pH (Guieu et al. 1994, Mackie, et al. 2005). At aerosol pH > 5, heterogeneous sulfate formation is dominated by the $O_3$ pathway because the solubility of $SO_2$ increases with increasing pH and because the S(IV) partitioning shifts to favor $SO_3^{2-}$ at higher pH. The aqueous-phase rate constant for $SO_3^{2-} + O_3$ is almost four orders of magnitude faster than for $HSO_3^- + O_3$ (Table 2). Heterogeneous sulfate formation by $NO_2$ also increases with increasing pH due to the increased solubility of $SO_2$ at high pH, and represents 15-30% of total heterogeneous sulfate production between pH = 6 – 8. In contrast to our results, previous results (Cheng et al., 2016; Wang et al., 2016) suggested that $NO_2$ oxidation is more important than $O_3$ oxidation at high pH values. The difference is due to assumed $O_3$ concentrations used in the rate calculations. The aforementioned studies assumed an $O_3$ concentration of 1 ppb relative to an $NO_2$ concentration of 66 ppb. In our model simulations, average $O_3$ and $NO_2$ concentrations are 9 ppb and 85 ppb, respectively. Modeled $O_3$ concentrations are similar to the observations during the measurement period (Figure S6). Heterogeneous sulfate production is lower (20%) when pH = 4 - 5 because of the low solubility of Fe and Mn and low concentrations of $SO_3^{2-}$ in this pH range. We note that while the pH-dependence of S(IV) partitioning is well known, the pH-dependency of metal solubility is more uncertain.

Anthropogenic $SO_2$ emissions in China have been reduced sharply since 2009 due to the stringent pollution control measures implemented (Zheng et al., 2018; Van der A et al., 2017; Krotkov et al., 2016). Compared with 2010, anthropogenic $SO_2$ emissions reduced by about 50% in 2015 (Krotkov et al., 2016; Zheng et al., 2018; Van der A et al., 2017). However, $NH_3$ and non-methane volatile organic compounds (NMVOC) emissions in China remained stable during 2010–2017 due to the absence of effective mitigation measures in current policies (Zheng et al., 2018). The emission changes may affect the

abundance of species that influence cloud and aerosol pH, and further influence sulfate production rates and the contribution of each sulfate formation pathway. However, other studies using observations between 2014-2016 (Liu et al., 2017; Song et al., 2018) found a similar pH range as calculated here, suggesting that a modeled low bias in aerosol pH is not likely to be the source of the modeled discrepancy in $\Delta^{17}O(SO_4^{2-})$.

Figure 5 also shows that model results with mean aerosol pH > 5 would result in a high bias in $\Delta^{17}O(SO_4^{2-})$ due to the increasing importance of the $O_3$ oxidation pathway at higher pH values, effectively providing an observational constraint for typical aerosol pH < 5 in Beijing haze. However, modeled $\Delta^{17}O(SO_4^{2-})$ calculated at the aerosol pH of 4.2-4.7 as estimated by recent studies (Guo et al. 2017; Song et al. 2018) represents a low bias relative to the observations. A joint comparison of observations and model results with both sulfate concentrations and $\Delta^{17}O(SO_4^{2-})$ suggests an average aerosol pH between 5-6 in Beijing during the study period, which is higher than bulk aerosol pH calculations in our model calculations and in previous studies (Guo et al. 2017; Song et al. 2018).

Thermodynamic pH calculations in ISORROPIA II are based on the assumption of internally mixed aerosols and do not directly consider aerosol alkalinity (carbonate). Fresh dust is initially alkaline (calcium carbonates). This alkalinity is partially depleted (carbonate is converted to $CO_2$) upon uptake of acid gases $HNO_3$, $SO_2$, and $H_2SO_4$ (Fairlie et al, 2010; Usher et al., 2003). Due to the high pH of alkaline dust, uptake of $SO_2$ would be followed by heterogeneous oxidation by $O_3$ (Fairlie et al., 2010; Ullerstam et al., 2003) and $NO_2$ (Zhao et al., 2018) if there is enough aerosol liquid water to promote aqueous-phase oxidation. A recent modeling study by Uno et al. (2017) using a previous version of GEOS-Chem explicitly includes uptake of $HNO_3$, $SO_2$, and $H_2SO_4$ on alkaline dust aerosols as described in Fairlie et al. (2010). Their model calculated that sulfate from uptake of $SO_2$ and $H_2SO_4$ on alkaline dust in Beijing during our measurement period represents on average 3% of total sulfate abundance. Most of this sulfate formation on alkaline dust was from uptake of $H_2SO_4$ (70 – 80%, Fairlie et al., 2010), with the remaining fraction (20 – 30%) from uptake of $SO_2$ followed by oxidation to sulfate. On average, less than 1% of sulfate in Beijing is formed on alkaline dust via the uptake and oxidation of $SO_2$ in the model.

At pH values representative of alkaline dust (pH = 7 – 8), sulfate formed via oxidation of $SO_2$ on alkaline dust would be dominated by $O_3$ leading to a relatively high $\Delta^{17}O(SO_4^{2-})$ value of 6‰ (Figure 5). In order to account for the difference in the median $\Delta^{17}O(SO_4^{2-})$ observed (1.0‰) and modeled (0.5‰), sulfate formation from the uptake and oxidation of $SO_2$ on alkaline dust would need to account for an average of 9% of total sulfate abundance during our measurement period. This fraction is higher than that calculated in Uno et al. (2017), which did not include anthropogenic dust. In our model, anthropogenic dust accounts for 28% of total dust in Beijing (Figure S7), and natural dust mostly originates from the Gobi Desert in southwestern Mongolia and the Badain Jaran Desert in northern China (Zhang et al., 2003; Zhang et al., 2016). The anthropogenic dust is not abundant enough to explain the difference between the results of Uno et al. (2017) and amount of sulfate production on alkaline dust required to explain the observed $\Delta^{17}O(SO_4^{2-})$. Due to uncertainties in processes such as

the RH dependence of the uptake of $SO_2$ on alkaline dust, the importance of this pathway could be underestimated in Uno et al. (2017). If this process does account for the difference in modeled and observed $\Delta^{17}O(SO_4^{2-})$, then $NO_2$ oxidation may be slightly more important than indicated in Figure 3. Our calculations (Figure 5) suggest that $NO_2$ oxidation accounts for 23% of heterogeneous sulfate formation at aerosol pH values between 7 and 8. If sulfate formation on alkaline dust accounts for 9% of total sulfate formation, and $NO_2$ oxidation accounts for 23% of this sulfate formation pathway, then $NO_2$ oxidation can account for up to 2% of total heterogeneous sulfate formation in Beijing, which still suggests $NO_2$ oxidation is a minor pathway during wintertime Beijing haze events.

In addition to a model underestimate of the ozone oxidation pathway, a model underestimate of the $H_2O_2$ oxidation pathway may explain part of the modeled low bias in $\Delta^{17}O(SO_4^{2-})$. The average modeled $H_2O_2$ concentrations during HPP and CP underestimates the observations (Ye et al., 2018) by up to an order of magnitude (Figure S3(c)). Mao et al., (2013) proposed a $HO_2-Cu-Fe$ catalytic mechanism for $H_2O_2$ production in the aerosol phase. In their mechanism, the uptake of $HO_2$ and subsequent heterogeneous reactions with Cu and Fe will lead to production of $H_2O_2$ when the molar ratio of dissolved Cu to Fe was >0.1. Ye et al. (2018) found that the molar ratio of dissolved Cu to Fe is >0.1 during moderately polluted days, suggesting the uptake of $HO_2$ radicals on particles might be an important source of $H_2O_2$ during the winter in Beijing. We note however that an underestimate of modeled $H_2O_2$ cannot explain all of the discrepancy in $\Delta^{17}O(SO_4^{2-})$, as sulfate formed from $H_2O_2$ oxidation (0.7 ‰) is lower than the observed mean $\Delta^{17}O(SO_4^{2-})$ of 0.9 ‰. All other oxidation pathways yield $\Delta^{17}O(SO_4^{2-}) = 0$ ‰ and cannot explain the model's low bias in $\Delta^{17}O(SO_4^{2-})$.

An increase in sulfate abundance of 9% from formation on alkaline dust is not enough to explain the remaining model underestimate in sulfate concentrations (-40%) and SOR (-28%) during HPP. Two recent studies suggested that hydroxymethane sulfonate (HMS) may be present in Chinese haze events and measured as sulfate via ion chromatography and HR-AMS (Moch et al., 2018; Song et al., 2019). HMS is formed via aqueous-phase oxidation via nucleophilic attack of $HSO_3^-$ and $SO_3^{2-}$ on HCHO. Moch et al. (2018) suggested that in-cloud formation of HMS could explain the low model bias in sulfate concentrations, while Song et al. (2019) suggested that heterogeneous formation of HMS on aerosol surfaces can explain up to one-third of the low model bias in their study. In the present study, as the organics were removed from the sample matrix prior to $\Delta^{17}O(SO_4^{2-})$ analysis (Geng et al., 2013), suggesting that measured $\Delta^{17}O(SO_4^{2-})$ does not have significant contributions from HMS. This will depend on the efficiency of HMS removal in He et al. (2018); previous work suggests that one of the two organic removal techniques employed ($H_2O_2$ oxidation) does not remove HMS under atmospheric conditions (Munger et al., 1984; 1986). However, the presence of HMS in the sample matrix, if any, would lower measured $\Delta^{17}O(SO_4^{2-})$, and any correction for this would increase the observed $\Delta^{17}O(SO_4^{2-})$ and further increase the discrepancy between modeled and observed $\Delta^{17}O(SO_4^{2-})$.

## 5. Conclusion

We have used a combination of observations and modeling of sulfate and $SO_2$ concentrations and $\Delta^{17}O(SO_4^{2-})$ to quantify sulfate production mechanisms in Beijing. We focus on the period of 17 October 2014 – 20 January 2015, when 10 heavy pollution periods (HPP) were defined with observed $PM_{2.5}$ concentrations >150 µg m$^{-3}$. The standard model simulation that only includes primary sulfate and sulfate formation from gas-phase oxidation by OH and in-cloud oxidation by $H_2O_2$ and $O_3$ underestimates mean sulfate concentration by 65% and $\Delta^{17}O(SO_4^{2-})$ by 50% during heavy pollution periods (HPP). Adding in-cloud oxidation catalyzed by transition metal ions (TMI) and heterogeneous oxidation by $H_2O_2$, $O_3$, $NO_2$, and TMI on aerosols can improve the model simulation of sulfate abundance and $\Delta^{17}O(SO_4^{2-})$, with the model biases decreasing from -65% to -40% for sulfate and from -50% to -5% for $\Delta^{17}O(SO_4^{2-})$ during HPP. Modeled heterogeneous sulfate production accounts for around 20% of total sulfate production. The model predicts that the TMI-catalyzed oxidation dominates heterogeneous sulfate production under calculated aerosol pH of $\leq 5$; however, this reaction is highly uncertain due to limitations in our ability to assess modeled dissolved Fe(III) and Mn(II) concentrations. The modeled $\Delta^{17}O(SO_4^{2-})$ is still biased low compared to observations, suggesting an underestimate of sulfate production by $O_3$ oxidation. We hypothesize that sulfate aerosol production by $O_3$ on externally-mixed alkaline dust aerosol can explain at least part of the remaining discrepancy in $\Delta^{17}O(SO_4^{2-})$. The $\Delta^{17}O(SO_4^{2-})$ observations suggest that a fractional sulfate contribution of just 9% originating from $SO_2$ oxidation on alkaline dust aerosol can explain the model discrepancy in $\Delta^{17}O(SO_4^{2-})$. We calculate that sulfate formation on alkaline dust is dominated by $O_3$ oxidation (74%) followed by $NO_2$ oxidation (23%). The $\Delta^{17}O(SO_4^{2-})$ observations combined with our model calculations indicates only a minor (2%) role of heterogeneous sulfate formation via $NO_2$ oxidation of $SO_2$. Future studies will examine the impact of these heterogeneous S(IV) oxidation mechanisms on the regional and global sulfur budgets.

**Data availability**

For the model results please contact Becky Alexander (beckya@uw.edu) and Lin Zhang (zhanglg@pku.edu.cn). For isotope measurements please contact Zhouqing Xie (zqxie@ustc.edu.cn)

**Author contribution**

BA, LZ, JYS, and YXW designed the study. BA and LZ supervised the project. JYS performed model simulations and conducted analyses with the assistance of QJC, YXW, XL, VS, RVM, SP, SJS and YZ. ZQX and PZH conducted the oxygen isotope measurements; YLS contributed the sulfate measurements. JYS, BA, and LZ wrote the paper. All authors contributed to the interpretation of the results and improvement of the paper.

**Acknowledgements**

This work is supported by the National Key Research and Development Program of China (2017YFC0210102) and the National Natural Science Foundation of China (41475112). J.Y. Shao acknowledges support from the Chinese Scholarship

490    Council (201606010049). B.A. Acknowledges support from NSF AGS 1644998 and Y. W. acknowledge support from NSF AGS 1645062. Z.Q.Xie acknowledges support from the National Natural Science Foundation of China (91544013). We thank Duncan Fairlie, Itushi Uno, Renyi Zhang, Havala O.T. Pye, Sakiko Ishino, Shohei Hattori, Youfan Chen, Jessica Haskins, Mi Zhou, Yuanhong Zhao for helpful discussions.

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

**Tables and Figures**

**Table 1.** Description of model simulations.

| Model run | TMI-catalysis in clouds | Heterogeneous reactions | Aerosol pH | Horizontal resolution |
|---|---|---|---|---|
| Run_STD | N | N | ISO[1] | 1/4°×5/16° & 4°×5° |
| Run_TMI | Y | N | ISO | 1/4°×5/16° & 4°×5° |
| Run_Het | Y | Y | ISO | 1/4°×5/16° & 4°×5° |
| Sensitivity simulations based on Run_Het[2] | | | | |
| Run_pH2 | Y | Y | 2 | 4°×5° |
| Run_pH3 | Y | Y | 3 | 4°×5° |
| Run_pH4 | Y | Y | 4 | 4°×5° |
| Run_pH5 | Y | Y | 5 | 4°×5° |
| Run_pH6 | Y | Y | 6 | 4°×5° |
| Run_pH7 | Y | Y | 7 | 4°×5° |
| Run_pH8 | Y | Y | 8 | 4°×5° |

[1] ISO means that the aerosol pH was calculated by ISORROPIA II in the model assuming metastable

 state.

[2] For the sensitivity simulations, aerosol pH was fixed to the stated prescribed value.

**Table 2** Aqueous-phase reaction rate expressions, rate constants ($k$) and influence of ionic strength ($I_s$) on the first-order aqueous-phase sulfate production in aerosol.

| Oxidants | $k$ (s$^{-1}$) | Uptake Gas | References |
|---|---|---|---|
| $O_3$ | $k_1[H_2SO_3]+k_2[HSO_3^-]+k_3[SO_3^{2-}]$ <br> $k_1 = 2.4 \times 10^4$ M$^{-1}$ s$^{-1}$ <br> $k_2 = 3.7 \times 10^5 \times \exp(-5530 \times (1/T-1/298))$ M$^{-1}$ s$^{-1}$ <br> $k_3 = 1.5 \times 10^9 \times \exp(-5280 \times (1/T-1/298))$ M$^{-1}$ s$^{-1}$ | $O_3$ | Hoffmann and Calvert (1985) |
| $H_2O_2$ | $k_4[H^+][HSO_3^-]/(1+K[H^+])$ <br> $k_4 = 7.45 \times 10^7 \times \exp(-4430 \times (1/T-1/298))$ M$^{-1}$ s$^{-1}$ <br> K = 13 M$^{-1}$ | $H_2O_2$ | McArdle and Hoffmann (1983) |
| $NO_2$ | $k_5[S(IV)]$ <br> $k_{5low} = 2 \times 10^6$ M$^{-1}$ s$^{-1}$ <br> $k_{5high} = (1.24 - 2.95) \times 10^7$ M$^{-1}$ s$^{-1}$ | $NO_2$ | Seinfeld and Pandis (2012) <br> Lee and Schwartz (1983) <br> Clifton, C. L. (1988) |
| TMI + $O_2$ | $k_6[H^+]^{-0.74}[Mn(II)][Fe(III)]$ (pH $\leqslant$ 4.2) <br> $k_6 = 3.72 \times 10^7 \times \exp(-8431.6 \times (1/T-1/297))$ M$^{-2}$ s$^{-1}$ <br> $k_7[H^+]^{0.67}[Mn(II)][Fe(III)]$ (pH > 4.2) <br> $k_7 = 2.51 \times 10^{13} \times \exp(-8431.6 \times (1/T-1/297))$ M$^{-2}$ s$^{-1}$ <br><br> $\log\left(\dfrac{k}{k^{I_s=0}}\right) = b_1\sqrt{I_s}/(1 + \sqrt{I_s})$ <br> $I_{s, max}$ = 2 M <br> $b_1$ is in range of -4 to -2 | $SO_2$ | Ibusuki and Takeuchi (1987) <br><br> Martin (1991) <br> Martin and Hill (1967) |
| HOBr | $k_8[HSO_3^-]+k_9[SO_3^{2-}]$ <br> $k_8 = 3.2 \times 10^9$ M$^{-1}$s$^{-1}$ <br> $k_9 = 5.0 \times 10^9$ M$^{-1}$s$^{-1}$ | HOBr | Chen et al. (2017) |
| $O_2$ on acidic microdroplet | $1.5 \times 10^6$ ( pH $\leq$ 3) | $SO_2$ | Hung and Hoffmann (2015) |

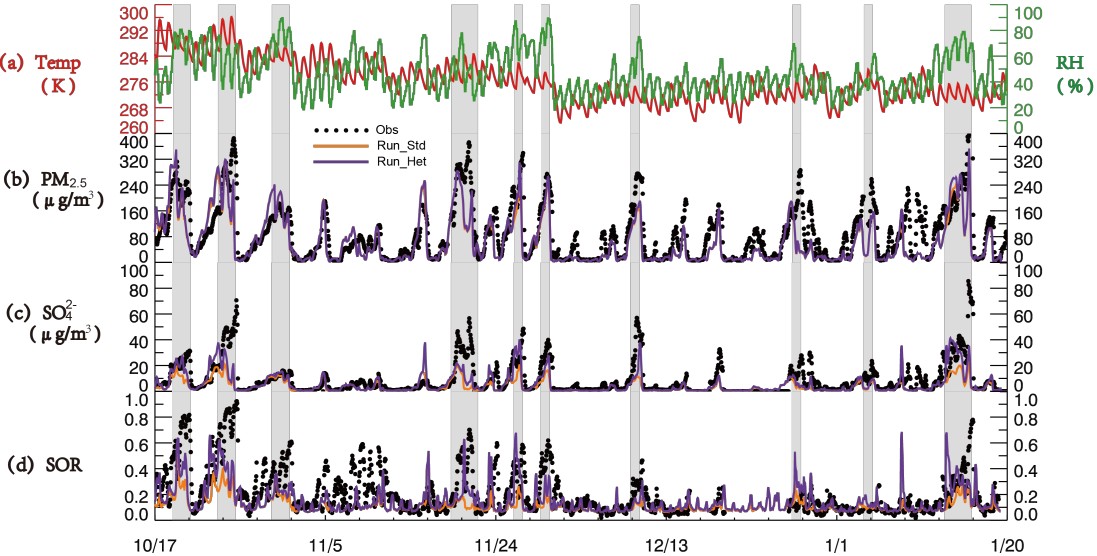

**Figure 1.** Time series of (a) GEOS-FP temperature (red line) and relative humidity (RH; green line), (b) PM$_{2.5}$ and (c) sulfate concentration, and (d) SOR at the surface in Beijing during the study period of 17 October 2014 – 20 January 2015. Hourly PM$_{2.5}$, sulfate, and SOR observations (black dots) are compared with model results from Run_Std (orange line) and Run_Het (purple line). The gray shaded bars represent 10 heavy pollution periods (HPP) (PM$_{2.5}$>150 µg m$^{-3}$) as defined in the text.

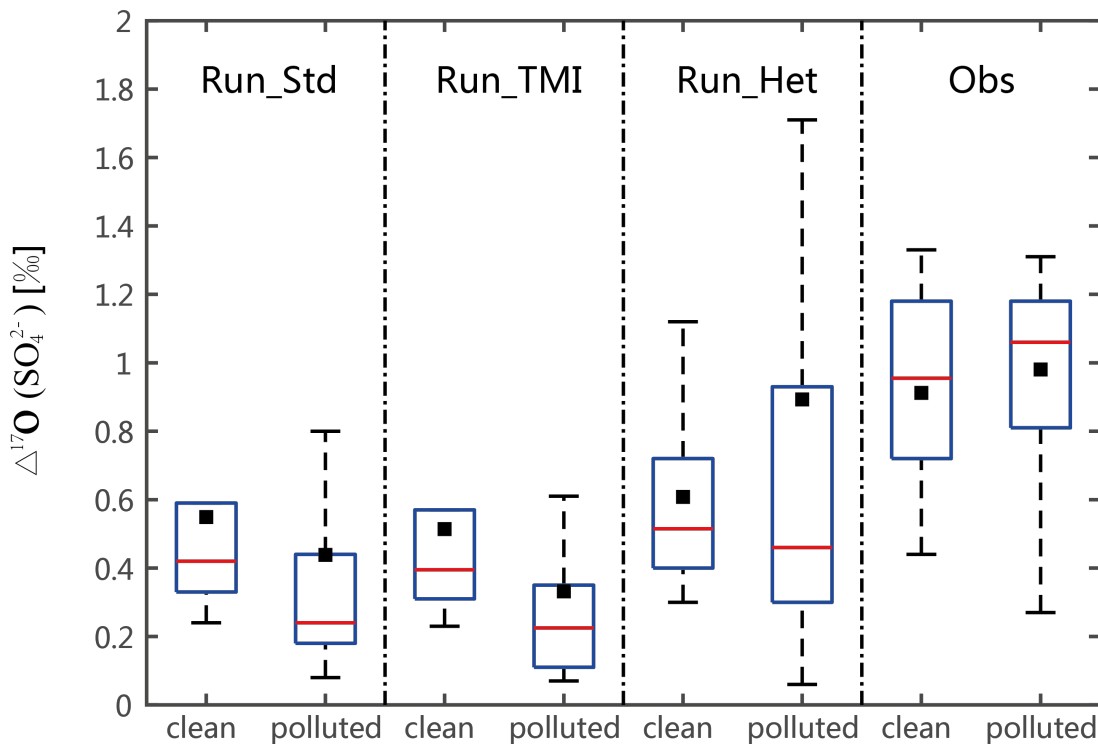

**Figure 2.** The box charts show observed vs. modeled $\Delta^{17}O(SO_4^{2-})$ in Beijing for each model
simulation separated for heavy pollution and clean periods. The box line from bottom to top
is respectively percentile of 25%, 50% and 75%, the whisker from bottom to top is respectively the
minimum and the maximum, and the black square is mean value.

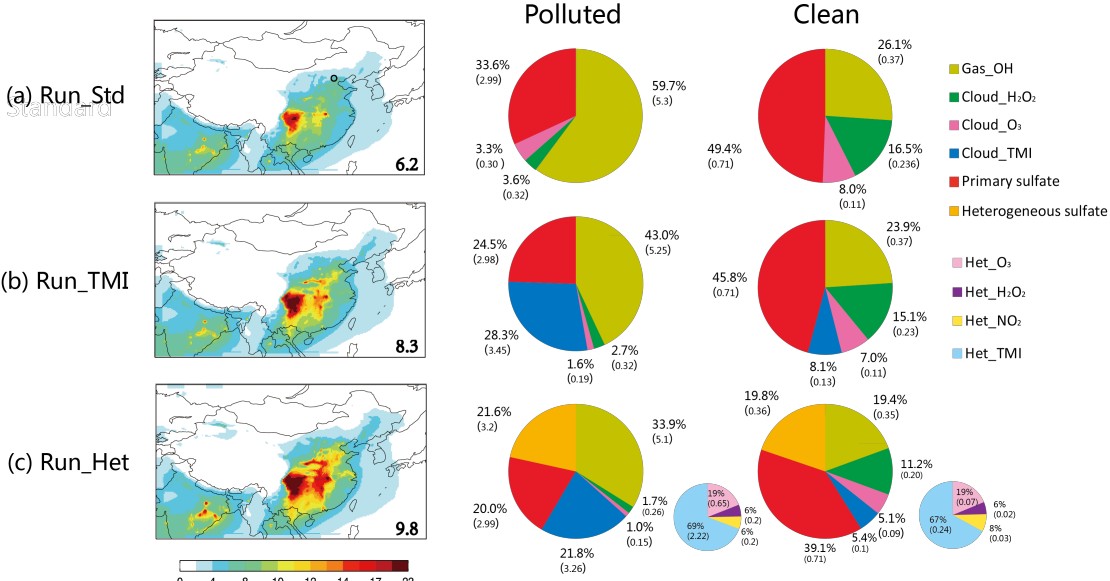

**Figure 3.** Model simulated sulfate aerosol concentrations (μg m$^{-3}$) above the ground for (a) Run_Std, (b) Run_TMI, and (c) Run_Het. Panels on the left show the spatial distributions with the numbers in inset representing simulated mean sulfate concentrations in Beijing (black circle in (a)) during the entire measurement period. The middle and right columns show percent contributions of different sulfate formation pathways to sulfate aerosol concentration in Beijing as calculated by the different model runs during polluted (HPP) and clean (CP) periods, respectively. The smaller pie charts in (c) show relative contributions of the four heterogeneous sulfate formation pathways implemented in the model. Numbers are percentage contributions (%) and absolute sulfate concentration (μg m$^{-3}$) are in parentheses.

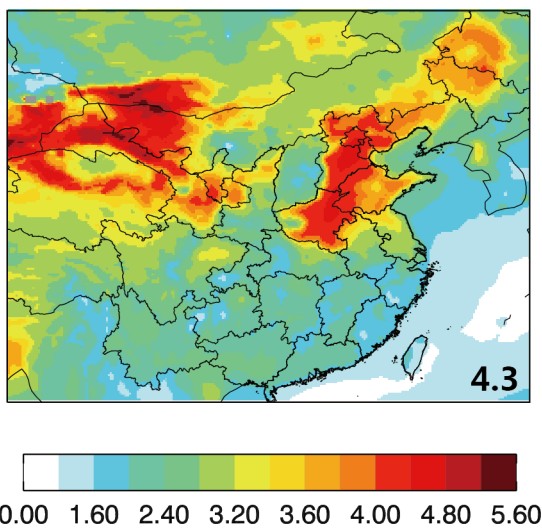

**Figure 4.** Model simulated mean aerosol pH values at the surface of China from 17 October 2014 to 20 January 2015 from Run_Het. Number in inset represents mean calculated pH values in Beijing.

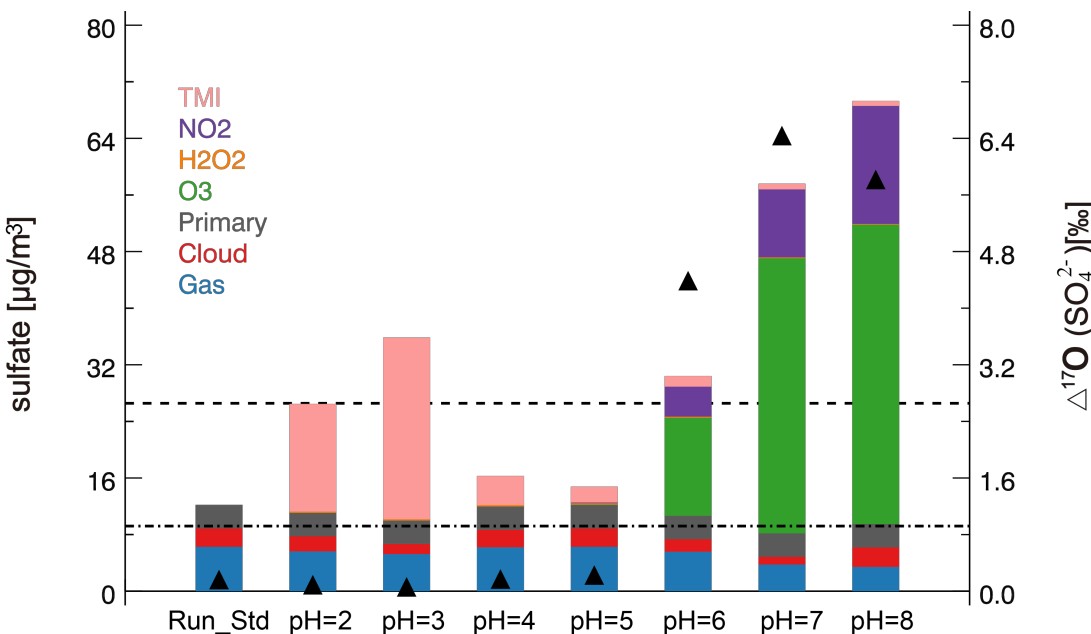

**Figure 5.** Model simulated mean sulfate concentration in Beijing averaged over heavy pollution periods (HPP as defined in the text). Model results are from the standard simulation (Run_Std) and sensitivity simulations with prescribed aerosol pH ranging 2 to 8. Different colors represent contributions from different sulfate formation pathways, including four heterogeneous reactions (oxidation by $O_2$ (TMI), $NO_2$, $H_2O_2$, $O_3$), primary anthropogenic, aqueous-phase oxidation in clouds, and gas-phase oxidation. Also shown are the corresponding modeled $\Delta^{17}O$ ($SO_4^{2-}$) values in Beijing (black triangles). The dashed line denotes observed mean sulfate concentration (25.9 ug m$^{-3}$), and the dot-dashed line denotes observed mean $\Delta^{17}O(SO_4^{2-})$ (0.9‰) during HPP in Beijing.