# Peer review of "Heterogeneous sulfate aerosol formation mechanisms during wintertime Chinese haze events: Air quality model assessment using observations of sulfate oxygen isotopes in Beijing"

_Atmospheric Chemistry and Physics, 2018_

## Referee Comment (RC1) · Anonymous Referee #1 · 12 Feb 2019

The study of Shao et al. Implemented four heterogeneous sulfate formation mechanism into GEOS-Chem model with a focus of the low biases of modeled sulfate production rates in China. The four paths are via H2O2, O3, NO2, and TMI on aerosols. In addition, TMI-catalyzed oxidation in clouds were also considered. To reduce the uncertainties, the oxygen isotopes observations were used for comparison as it's highly sensitive to the relative importance of different sulfate production mechanism. To investigate the dependence on aerosol PH, sensitivity studies with prescribed values of aerosol PH were also conducted. The design of the experiments are comprehen-

sive and the results are convincing. Overall, the results with the four added heterogeneous reactions significantly reduced the low biases of sulfate and also oxygen isotopes, which show better agreement with the observations. The publication is very well written, clearly structured, and the analyses are comprehensive and convincing. In particular, the authors present a thorough analysis of the heterogeneous oxidation paths, substituting the bulk first-order uptake of SO2 (reaction probability /uptake coefficients) by a more specific calculation approach. The paper has a good chance to become an important reference for future studies on the sulfate heterogeneous reaction mechanism in China. It also has regional/global impacts. I support publication of this manuscript and have only a few small comments that may require minor revision.

1. P5 L45 Can you add some more details of the methodology of the in-cloud TMI-catalyzed aqueous-phase S(IV) oxidation? As it seemed this in-cloud TIM-relevant path is very important in polluted events.

2. P5 L45 The TMI-catalyzed oxidation path are important both in-cloud (aqueous) and on-aerosol (heterogeneous). Is there any possibility to verify the assumption of Fe and Mn treatments in the model, from the natural dust and anthropogenic emissions, solubility, ionic strength of cloud liquid water, etc.? Or if not, is it necessary to add some discussions about the uncertainty of those assumptions and the possibility of the impacts on the analysis?

3. P7 L110 The 2010_MEIC emission is used in the study, however the anthropogenic SO2 emissions has been reduced largely since 2009 thus the SO2 overestimation might be expected for Beijing. Is there any impacts/uncertainty induced by this issue?

In the Run_Het results, the TMI-relevant reactions are the most important among the four heterogeneous paths during both clean and polluted events. "In the model, the heterogeneous sulfate production rate from the TMI-catalyzed reaction is calculated as first-order uptake in SO2". Is that relevant with the SO2 overestimation in the model?

4. P7 L120 The mass accommodation coefficients for O3 andNo2 are much lower that

for SO2. Is this the reason that TMI-relevant reactions are much important in the analysis? How those parameters were determined? Any assumption? Is it necessary to put Text S2 in the manuscript? As the analysis is done for the October-November period, is it possible to apply to other seasons? any sensitivity/uncertainties to regions/seasons etc?

5. P11 L345 It seemed the sulfate bias is reduced while the PM2.5 bias is still there. I'm just wondering is there any observational data of nitrate or ammonia available for comparison? It might give some indication of the partitioning of sulfate-nitrate. If the nitrate are well simulated or not impacted by the competition from sulfate formation, larger improvements of PM2.5 are expected?

6. P11 L349 The largest sulfate enhancements due to heterogeneous sulfate formation occur in megacities in eastern China and Sichuan Basin. For Sichuan region, is that also due to the high cloud liquid water path/RH?

7. In Run_HET, the simulated oxygen isotopes are improved in average but the median are largely underestimated. In addition to the assumption of O3 oxidation underestimation, is there any possibility of other missing paths that not taken into account? Any discussions necessary?

---

## Referee Comment (RC2) · Anonymous Referee #2 · 17 Feb 2019

It has been a long question regarding the origin(s) of the non-zero $\Delta17O$ (but also $\Delta33S$ and $\Delta36S$ values) observed in tropospheric aerosols. This study implements the four majors SO2 oxidation pathways into the GEOS-Chem model in clouds and on aerosols and compared with the sulfates measured in China (coupled with the O-isotopes). This study also discussed the contribution of the different oxidation pathways by estimating the pH in urban aerosols. While current models underestimate the sulfate formation in aerosols, results in this study significantly reduced the difference between observed and modeled in both sulfates concentrations and oxygen isotopes.

[Figure]

However uncertainties remain but it has been identified that O2+TMI oxidation pathway is dominant when pH < 5. Another oxidation pathways on alkaline dust has also been hypothetized (very speculative) to explain the high $\Delta$ 17O-values.

General comments:

1) Input of stratospheric sulfates and sulfates resulting from Criegee radicals oxidation pathway are nowadays largely discussed when taking into account the S-isotopes. It is suggested that those processes are responsible for high $\Delta$ 33S-values which cannot be explained yet by current knowledge. How would those processes affect the results ? Is it possible to determine their contribution ? I suggest these processes being discussed.

2) The std_run model can account for the seasonality observed in the aerosols. Does the implementation of the four oxidation pathways change the results ? Is it more accurate ? If not, what would be the drivers ?

3) The RHs seem to be an important factor when it comes to understand the sulfate formation. However, as described in the paper, whywould the Sichuan province be the only one to exhibit the largest enhancement of sulfate formation ? Many places in China might be characterized by high humidity and air stagnation ?

4) The model supposes a constant contribution of primary aerosols emitted by the human. However the contribution of each sources might change depending of the seasons which should also affect the O-isotopes (towards 0 ‰o). Thus how would this sources variations change the results ?

5) Recent studies suggested that dust particles would be responsible for pollution in China during winter, this certaintly due to photooxidation by mineral dust. Does the new models takes into account input of dust mineral from desert etc ? It has also been shown that others reactions occurred on the dust particles besides NO2 and ozone (OH for example). How would it affect the O-isotopes ? What would be the contribution

of this oxidation pathway during the winter ? Would it change the contribution value (9%) ? I suggested this being discussed.

6) Although this study is very interesting and promising, results show that many uncertainties remain when it comes to understand the different oxidation pathways undertaken by SO2. I suggest that the conclusion should be more contrasted as O2+TMI cannot alone account for all the O-isotopes signatures.

7) With all the pollution in China, none of the models predict high pollution in Beijing which is surprising. How could it be explained ?

8) In Figure 1, why the SOR is not correlated to SO4 during the 11/5 ?

---

## Author Comment (AC1) · 6 Apr 2019

Reviewer #1

The study of Shao et al. Implemented four heterogeneous sulfate formation mechanism into GEOS-Chem model with a focus of the low biases of modeled sulfate production rates in China. The four paths are via H2O2, O3, NO2, and TMI on aerosols. In addition, TMI-catalyzed oxidation in clouds were also considered. To reduce the uncertainties, the oxygen isotopes observations were used for comparison as it's highly sensitive to the relative importance of different sulfate production mechanism. To investigate the dependence on aerosol PH, sensitivity studies with prescribed values of aerosol PH were also conducted. The design of the experiments are comprehensive and the results are convincing. Overall, the results with the four added heterogeneous reactions significantly reduced the low biases of sulfate and also oxygen isotopes, which show better agreement with the observations. The publication is very well written, clearly structured, and the analyses are comprehensive and convincing. In particular, the authors present a thorough analysis of the heterogeneous oxidation paths, substituting the bulk first-order uptake of SO2 (reaction probability /uptake coefficients) by a more specific calculation approach. The paper has a good chance to become an important reference for future studies on the sulfate heterogeneous reaction mechanism in China. It also has regional/global impacts. I support publication of this manuscript and have only a few small comments that may require minor revision.

**Response: We thank the reviewer for the valuable comments. All of them have been implemented in the revised manuscript. Please see our itemized response below.**

**Comment:** 1. P5 L45 Can you add some more details of the methodology of the in-cloud TMI- catalyzed aqueous-phase S(IV) oxidation? As it seemed this in-cloud TMI-relevant path is very important in polluted events.

**Response: Thanks for pointing it out. Metal-catalyzed S(IV) oxidation by $O_2$ (TMI) in cloud droplets was considered to be one of the most important sulfate formation pathways during the winter in the North Hemisphere (Huang et al., 2014; Harris et al., 2013; Alexander et al., 2009; Sofen et al., 2011), and we also found that the pathway accounted for over 20% of total sulfate formation during HPP at Beijing (P11L355). We now state in the text (P5L145) "In the model simulation Run_TMI, we implemented the in-cloud TMI-catalyzed aqueous-phase S(IV) oxidation by $O_2$ into the model, which is thought to be one of the most important sulfate formation pathways during the North Hemisphere winter (Huang et al., 2014; Harris et al., 2013; Alexander et al., 2009; Sofen et al., 2011). The parameterization of TMI-catalyzed S(IV) oxidation in cloud for GEOS-Chem follows Alexander et al. (2009)."**

**We have also added more details on the methodology of the in-cloud TMI-catalyzed aqueous-phase S(IV) oxidation in the Supplement (Text S3; Figure S4; Table S2; Table S3). This also addresses Comment #2 below. We now state in the text (P5L155) "After modification, the average aqueous-phase concentration of Fe(III) in cloud water during our studying period is 2.9 μM and Mn(II) is 1.3 μM in the model, which is consistent with previous work (He et al., 2018; Shen et al., 2012; Guo et al., 2012) (see Text S3 in the Supplement for more details)."**

**Text S3 in the Supplement is as follows:**

The parameterization of TMI-catalyzed S(IV) oxidation in clouds in GEOS-Chem is described in Alexander et al., (2009). The natural source of Fe ($[Fe]_{nat}$) and Mn ($[Mn]_{nat}$) from mineral dust are scaled to total dust mass. $[Fe]_{nat}$ is 3.5% of total dust mass and $[Mn]_{nat}$ is a factor of 50 lower than $[Fe]_{nat}$ (Alexander et al., 2009). The anthropogenic source of Fe ($[Fe]_{ant}$) and Mn ($[Mn]_{ant}$) from coal combustion are scaled to the abundance of primary anthropogenic sulfate due to their common source and atmospheric lifetime $[Mn]_{ant}$ is 1/300 of primary sulfate concentration and $[Fe]_{ant}$ is 10 times that of $[Mn]_{ant}$ as described in Alexander et al. (2009). Figure S4 show the distribution of simulated $[Fe]_{nat}$ and $[Fe]_{ant}$ during the entire model simulation period. Table S2 compares autumn-winter mean modeled Mn and Fe with the observations at several sites in east Asia, including 3 sites in the North China Plain (NCP) region (Beijing, Tianjin and Shijiazhuang), 2 sites in the Sichuan Basin (Chengdu and Chongqing), and Lanzhou and Seoul which are the upwind and downwind of Beijing, respectively. The model tends to underestimate both Fe and Mn at the NCP sites by a factor of 0.25-0.72. Model comparison with observations in the Sichuan Basin shows good agreement on Mn concentration (within 20% for both Chongqing and Chengdu), with model calculations overestimating the observations of Fe concentrations. For sites in Lanzhou (northwest of Beijing, upwind) and Seoul (southeast of Beijing, downwind), model comparison with observations indicates good agreement on Fe (within 5%), and underestimates Mn by 60%. The model underestimates Fe and Mn in the NCP region and overestimates in Sichuan Basin. These discrepancies highlight the limitations of our approach applying a global-scale factor of $[SO4]_{primary}/[Mn]_{ant}$ and $[dust]/[Fe]_{nat}$, as regionally varying emissions control technologies and mass fraction of Fe in dust may impact the relative emission rates.

However, sulfate formation by the TMI-catalyzed oxidation pathway is influenced by soluble $Fe^{3+}$ and $Mn^{2+}$ concentrations, as opposed to total Fe and Mn. Previous studies suggest that the solubility of Fe and Mn ranges from 0.03 to 54% and 1.2 to 97%, respectively. The solubility of metals is influenced by several factors such as the natural versus anthropogenic origin of samples (A.R. Baker et al., 2006; K.V. Desboeufs et al., 2005; K.V. Desboeufs et al., 2001; Spokes et al., 1994; A. Ito1 and Y. Feng, 2010; P.Y. Chuang et al., 2005; Solmon et al., 2009), acidity, and sunlight.  In this study, we assume a solubility of 10% for $[Fe]_{ant}$, 0.45% for $[Fe]_{nat}$, 50% for $[Mn]_{ant}$, and 5% for $[Mn]_{nat}$ in cloud water. The modeled soluble $Fe^{3+}$ and $Mn^{2+}$ concentration is shown in Table S3. After modification, the average modeled concentration of soluble Fe(III) in cloud water during our study period is $2.9\pm1.8$ μM and Mn(II) is $1.3\pm0.7$ μM, which is consistent with estimates (He et al., 2018) and observations (Shen et al., 2012; Guo et al., 2012). A sensitivity study is performed based on Run_TMI but with the higher solubility of Fe and Mn as Alexander et al. (2009): the solubility of 1% of $[Fe]_{nat}$ and 50% for $[Mn]_{nat}$ in cloud water. The modeled soluble Fe(III) and Mn(II) concentration reaches 20μM and 10 μM during HPP, up to a factor of 5 higher than the observations. The simulated sulfate concentration is also overestimated the observation by around 100% during HPP in Beijing.

[Figure]

**Figure S4.** Modeled anthropogenic and natural Fe and Mn (ng m$^{-3}$) at the surface, along with the anthropogenic/total Fe and Mn concentration percentages.

**Table S2.** Comparison of observed and modeled autumn and winter mean Mn and Fe concentrations (ng m$^{-3}$) from several East Asian locations.

| Location | Fe (Mn) observation (ng m$^{-3}$) | Fe (Mn) model (ng m$^{-3}$) | Fe (Mn) Model/observation | Reference |
|---|---|---|---|---|
| Beijing | 1800 (90) | 835 (22.7) | 0.46 (0.25) | Zhao et al. (2013) |
| Tianjin | 1980 (120) | 1239 (36.2) | 0.72 (0.30) | |
| Shijiazhuang | 2250 (150) | 1615 (44.3) | 0.63 (0.30) | |
| Chengdu | 875 (38) | 1245.7 (34.1) | 1.42 (0.90) | Wang et al. (2018) |
| Chongqing | 502.5 (36.5) | 1226.0 (39.7) | 2.4 (1.09) | |
| Lanzhou | 1534 (59) | 1456.9 (33.0) | 0.95 (0.56) | Wang et al. (2016) |
| Seoul | 227 (21) | 219.8 (6.1) | 0.97 (0.30) | Park et al. (2018) |

**Table S3.** Comparison of observed and modeled soluble Mn(II) and Fe(III) concentrations (μM).

| | This study Range (monthly mean) | Beijing (He et al., 2018) | Mt. Tai[a] (Shen et al., 2012) | Mt. Tai (Guo et al., 2012) |
|---|---|---|---|---|
| Fe(III) | 0.4-11.2 (2.9±2.7) | 0.6-6.1 (2.6±1.8) | 0.8-7.4 | 2.6 |
| Mn(II) | 0.4-4.8 (1.3±0.9) | 1 | 0.4-1.7 | 1.2 |

[a]Mt. Tai is located in central Shandong province at the eastern edge of the NCP region.

**Comment:** 2. P5 L45 The TMI-catalyzed oxidation path are important both in-cloud (aqueous) and on-aerosol (heterogeneous). Is there any possibility to verify the assumption of Fe and Mn treatments in the

model, from the natural dust and anthropogenic emissions, solubility, ionic strength of cloud liquid water, etc.? Or if not, is it necessary to add some discussions about the uncertainty of those assumptions and the possibility of the impacts on the analysis?

**Response: Thanks for pointing it out. We now state in the text (P5L155): "After modification, the average modeled concentration of Fe(III) in cloud water during our study period is 2.9 μM and that of Mn(II) is 1.3 μM, which are consistent with previous work (He et al., 2018; Shen et al., 2012; Guo et al., 2012) (see Text S3 in the Supplement for more details)." Natural and anthropogenic dust emissions are now shown in Figure S7. The influence of ionic strength on the in-cloud, TMI-catalyzed S(IV) oxidation pathway is negligible, but is significant in aerosol water (heterogeneous pathway). Please refer to the response for Comment #1 and Text S3 in the Supplement for the discussions on Fe and Mn treatments and associated uncertainties.**

**In addition, we deleted the sentence on P5L155 to avoid misunderstanding. We also rewrite the sentence on P6L165 "In addition, we considered the impacts of acidity and ionic strength on TMI-catalyzed reaction rates following Cheng et al. (2016) (Table S2 in the SI of Cheng et al., 2016), since the ionic strength of aerosol liquid water can reach 20 M during polluted periods (He et al., 2018; Herrmann et al., 2015)."**

**Comment:** 3. P7 L110 The 2010_MEIC emission is used in the study, however the anthropogenic SO2 emissions has been reduced largely since 2009 thus the SO2 overestimation might be expected for Beijing. Is there any impacts/uncertainty induced by this issue?

In the Run_Het results, the TMI-relevant reactions are the most important among the four heterogeneous paths during both clean and polluted events. "In the model, the heterogeneous sulfate production rate from the TMI-catalyzed reaction is calculated as first-order uptake in SO2". Is that relevant with the SO2 overestimation in the model?

**Response: Thanks for pointing this out. Unfortunately, the updated MEIC emission inventory was not released at the time when we ran our model simulations. We now state in the text (P13L388) "Anthropogenic SO₂ emissions in China have been reduced sharply since 2009 due to the stringent pollution control measures implemented (Zheng et al., 2018; Van der A et al., 2017; Krotkov et al., 2016). Compared with 2010, anthropogenic SO₂ emissions reduced by about 50% in 2015 (Krotkov et al., 2016; Zheng et al., 2018; Van der A et al., 2017). NH₃ and non-methane volatile organic compounds (NMVOC) emissions in China remained stable during 2010–2017 due to the absence of effective mitigation measures in current policies (Zheng et al., 2018). The emission changes may affect the abundances of species that influence cloud and aerosol pH, and further influence sulfate production rates and the contribution of each sulfate formation pathway. However, other studies using observations between 2014-2016 (Liu et al., 2017; Song et al., 2018a) found a similar pH range as calculated here, suggesting that a modeled low bias in aerosol pH is not likely to be the source of the modeled discrepancy in $\Delta^{17}O(SO_4^{2-})$."**

**We believe that the SO₂ overestimation in the model will not change the dominance of TMI-relevant pathway in the model because SO₂ concentrations influence all four heterogeneous sulfate**

formation pathways. As shown in Figure S1, heterogeneous sulfate production rates parameterized as first-order in $SO_2$ or first-order in the oxidant result in similar values when aerosol pH < 6, the only difference is when production rates are limited by the mass transport across the air-water interface at higher pH values. $SO_2$ concentration influence the sulfate formation by gas-phase oxidation by OH and aqueous-phase oxidation by $H_2O_2$, $O_3$ and $O_2$ catalyzed oxidation by TMI. Anthropogenic sulfate emission also scales to anthropogenic $SO_2$ emission in the model. Therefore, the $SO_2$ overestimation would not change the main conclusion of this study regarding the fractional contribution of each sulfate formation pathway to total sulfate abundance in Beijing.

**Comments:** 4. P7 L120 The mass accommodation coefficients for O3 and NO2 are much lower that for SO2. Is this the reason that TMI-relevant reactions are much important in the analysis? How those parameters were determined? Any assumption? Is it necessary to put Text S2 in the manuscript? As the analysis is done for the October-November period, is it possible to apply to other seasons? any sensitivity/uncertainties to regions/seasons etc?

**Response: Thanks for suggestions. The mass accommodation coefficients would influence heterogeneous sulfate production rates when aerosol pH >6, but the aerosol pH seldom reaches these values in the model. Even if aerosol pH reaches 6 in the model, $Fe^{3+}$ and $Mn^{2+}$ concentrations are very low due to low solubility at high pH (Cheng et al., 2016), thus less sulfate would be produced by this pathway under this condition. Therefore, the mass accommodation coefficients are not the reason that TMI-relevant reactions are so important among the four heterogeneous pathways. The mass accommodation coefficients for $O_3$, $NO_2$, $H_2O_2$ and $SO_2$ are from laboratory calculations and widely used (Jacob, 2000). Considering the importance of Table S2, we now move it to the manuscript.**

**The importance of heterogeneous sulfate formation for the global sulfur budget and in different seasons is the focus of our next paper, and is beyond the scope of this manuscript.**

**Comments:** 5. P11 L345 It seemed the sulfate bias is reduced while the PM2.5 bias is still there. I'm just wondering is there any observational data of nitrate or ammonia available for comparison? It might give some indication of the partitioning of sulfate-nitrate. If the nitrate are well simulated or not impacted by the competition from sulfate formation, larger improvements of PM2.5 are expected?

**Response:**

**Thanks for the comment. Both the sulfate and $PM_{2.5}$ low biases still exist, even after implementing the four heterogeneous sulfate production pathways, as discussed in the manuscript. We have discussed in the text the potential influence of HMS in reconciling the remaining low bias in sulfate. This paper focuses on sulfate production mechanisms. Investigation of model biases in nitrate and ammonia are outside the scope of this paper, although we do have a separate paper in preparation focusing on a comparison of modeled and observed nitrate in Beijing. In the figure below, we compare modeled and observed nitrate and ammonia concentrations, which shows that the enhancements in sulfate abundance in Run_Het result in a small increase in ammonia, but have a negligible effect on nitrate concentrations. As noted in previous work (Wang et al., 2013), the model**

**tends to overestimate nitrate concentrations.**

[Figure]

**Figure.** Time series of nitrate and ammonium at the surface in Beijing during the study period of 17 October 2014 – 20 January 2015. 12-hour average nitrate and ammonium observations (black line) are compared with model results from Run_Std (orange line) and Run_Het (purple line).

**Comments:** 6. P11 L349 The largest sulfate enhancements due to heterogeneous sulfate formation occur in megacities in eastern China and Sichuan Basin. For Sichuan region, is that also due to the high cloud liquid water path/RH?

**Response: Thanks for pointing this out. We now state in the text (P10L320): "The largest enhancement in sulfate abundance after adding the in-cloud TMI pathway occurs in Sichuan basin (around 6.5 µg m$^{-3}$), where simulated anthropogenic Fe and Mn from coal fly ash (Figure S4) and SO$_2$ are high (Zhang et al., 2009) in part due to high SO$_2$ emissions combined with stagnant air and high relative humidity all year (Huang et al., 2014)." We have also compared the observed and modeled Fe and Mn concentrations in the Sichuan region as shown in Table S2.**

**Comments:** 7. In Run_HET, the simulated oxygen isotopes are improved in average but the median are largely underestimated. In addition to the assumption of O3 oxidation underestimation, is there any possibility of other missing paths that not taken into account? Any discussions necessary?

**Response: Ozone is the only oxidant with a $\Delta^{17}O$ value high enough to explain the majority of the discrepancy. The ozone oxidation pathway is mainly limited by the fraction of S(IV) that exists as SO$_3^{2-}$, which increases with increasing aerosol pH. In the manuscript, we have suggested that sulfate formed via oxidation of SO$_2$ by ozone on alkaline dust could explain at least part of the discrepancy, as this reaction is not included in the model (P14L440).**

**An underestimate of H$_2$O$_2$ could also explain part of the discrepancy. We now state in the text "The average modeled H$_2$O$_2$ concentrations during HPP and CP (Figure S3) underestimates the observations (Ye et al., 2018) by up to an order of magnitude (Figure S3). Mao et al. (2013) proposed a HO$_2$−Cu−Fe catalytic mechanism for H$_2$O$_2$ production in the aerosol phase. In their mechanism, the uptake of HO$_2$ and subsequent heterogeneous reactions with Cu and Fe will lead to production of H$_2$O$_2$ when the molar ratio of dissolved Cu to Fe was >0.1. Ye et al. (2018) found that the molar ratio of dissolved Cu to Fe could be >0.1 during moderately polluted days, suggesting the uptake**

of $HO_2$ radicals on particles might be an important source of $H_2O_2$ during the winter in Beijing. We note however that an underestimate of modeled $H_2O_2$ cannot explain all of the discrepancy in $\Delta^{17}O(SO_4^{2-})$, as sulfate formed from $H_2O_2$ oxidation (0.7 ‰) is lower than the observed mean $\Delta^{17}O(SO_4^{2-})$ of 0.9 ‰. All other oxidation pathways yield $\Delta^{17}O(SO_4^{2-}) = 0$ ‰ and cannot explain the model's low bias in $\Delta^{17}O(SO_4^{2-})$."

---

## Author Comment (AC2) · 6 Apr 2019

Reviewer #2

It has been a long question regarding the origin(s) of the non-zero $\Delta 17O$ (but also $\Delta 33S$ and $\Delta 36S$ values) observed in tropospheric aerosols. This study implements the four majors SO2 oxidation pathways into the GEOS-Chem model in clouds and on aerosols and compared with the sulfates measured in China (coupled with the O- isotopes). This study also discussed the contribution of the different oxidation path- ways by estimating the pH in urban aerosols. While current models underestimate the sulfate formation in aerosols, results in this study significantly reduced the difference between observed and modeled in both sulfates concentrations and oxygen isotopes. However uncertainties remain but it has been identified that O2+TMI oxidation pathway is dominant when pH < 5. Another oxidation pathways on alkaline dust has also been hypothetized (very speculative) to explain the high $\Delta 17O$-values.

**Response: We thank the reviewer for the valuable comments. All of them have been implemented in the revised manuscript. Please see our itemized response below.**

**Comments: 1.** Input of stratospheric sulfates and sulfates resulting from Criegee radicals oxidation pathway are nowadays largely discussed when taking into account the S-isotopes. It is suggested that those processes are responsible for high $\Delta 33S$-values which cannot be explained yet by current knowledge. How would those processes affect the results? Is it possible to determine their contribution? I suggest these processes being discussed.

**Response: Thanks for pointing this out. Based on the theoretical mechanism of sulfate production from the gas-phase oxidation of SO$_2$ by stabilized Criegee Intermediats (sCI) described in Taatjes et al. (2014), the $\Delta^{17}O$ value of sulfate formed through this mechanism will be equal to 0 permil as it is the central oxygen atom of ozone is transferred to the sulfate product. Additionally, it is expected that sCIs are not important for sulfate formation in wintertime in Beijing (Pierce et al., 2013). We have reiterated in the discussion (P14L439) that all sulfate formation pathways other than ozone and H$_2$O$_2$ cannot explain the modeled low bias in $\Delta^{17}O(SO_4^{2-})$ as they produce sulfate with $\Delta^{17}O(SO_4^{2-})$ = 0‰. Given that the HPP periods occur during time periods of stable conditions and a stagnant boundary layer with little vertical mixing, the stratospheric contribution at the surface is likely minor.**

**Taatjes, C. A., Shallcross, D. E., and Percival, C. J.: Research frontiers in the chemistry of Criegee intermediates and tropospheric ozonolysis, Phys Chem Chem Phys, 16, 1704-1718, doi:10.1039/c3cp52842a, 2014.**

**Pierce, J. R., Evans, M. J., Scott, C. E., Andrea, S. D. D', Farmer, D. K., Swietlicki, E., and Spracklen, D. V.: Weak global sensitivity of cloud condensation nuclei and the aerosol indirect effect to Criegee + SO$_2$; chemistry, Atmos. Chem. Phys., 13, 3163-3176, doi:10.5194/acp-13-3163-2013, 2013.**

**Comments: 2.** The std_run model can account for the seasonality observed in the aerosols. Does the implementation of the four oxidation pathways change the results? Is it more accurate? If not, what would be the drivers?

**Response: Thanks for this suggestion. The importance of four heterogeneous sulfate formation for global and seasonal variation is the focus of our next paper, and is beyond the scope of the current manuscript. In the current manuscript, we focus on autumn and winter in Beijing when $PM_{2.5}$ concentrations are highest and $\Delta^{17}O(SO_4^{2-})$ observations are available at relatively high time resolution.**

**Comments: 3.** The RHs seem to be an important factor when it comes to understand the sulfate formation. However, as described in the paper, why would the Sichuan province be the only one to exhibit the largest enhancement of sulfate formation? Many places in China might be characterized by high humidity and air stagnation?

**Response: Thanks for pointing this out. We now state in the text (P10L320): "The largest enhancement in sulfate abundance after adding the in-cloud TMI pathway occurs in Sichuan basin (around 6.5 μg m$^{-3}$), where simulated anthropogenic Fe and Mn from coal fly ash (Figure S4) and $SO_2$ are high largely due to high $SO_2$ emissions (Zhang et al., 2009) combined with stagnant air and high relative humidity all year (Huang et al., 2014)."**

**Comments: 4.** The model supposes a constant contribution of primary aerosols emitted by the human. However the contribution of each sources might change depending of the seasons which should also affect the O-isotopes (towards 0 ‰). Thus how would this sources variations change the results?

**Response: Thanks for this comment. The importance of four heterogeneous sulfate formation for sulfate seasonal variation is the focus of our next paper, and is beyond the scope of the current manuscript.**

**Comments: 5.** Recent studies suggested that dust particles would be responsible for pollution in China during winter, this certaintly due to photooxidation by mineral dust. Does the new models takes into account input of dust mineral from desert etc? It has also been shown that others reactions occurred on the dust particles besides NO2 and ozone (OH for example). How would it affect the O-isotopes? What would be the contribution of this oxidation pathway during the winter? Would it change the contribution value (9%)? I suggested this being discussed.

**Response: Thanks for this suggestion. Run_Het does account for desert dust. We now state in the text (P13L425) "Modeled anthropogenic dust accounts for 28% of total dust in Beijing (Figure S7), and natural dust mostly originates from the Gobi Desert in southwestern Mongolia and the Badain Jaran Desert in northern China (Zhang et al., 2003; Zhang et al., 2016). The anthropogenic dust is not abundant enough to explain the difference between the results of Uno et al. (2017) and amount of sulfate production on alkaline dust required to explain the observed $\Delta^{17}O(SO_4^{2-})$." We now present the natural and anthropogenic dust distributions in China in Figure S7.**

[Figure]

**Figure S7.** Modeled anthropogenic and natural dust (μg m⁻³) at the surface, along with the percentage contribution of anthropogenic dust to total dust concentration.

**Comments: 6.** Although this study is very interesting and promising, results show that many uncertainties remain when it comes to understand the different oxidation pathways undertaken by SO2. I suggest that the conclusion should be more contrasted as O2+TMI cannot alone account for all the O-isotopes signatures.

**Response: We agree that there are many remaining uncertainties, and that the TMI pathway is particularly uncertain due to limitations in the ability to model soluble Fe(III) and Mn(II). A discussion of this limitation is in Section Text S3. In the conclusions, we now emphasize that these results are a model prediction that is highly uncertain (P15L470): "The model predicts that TMI-catalyzed oxidation dominates heterogeneous sulfate production under calculated aerosol pH of ≤5; however, this reaction is highly uncertain due to limitations in our ability to assess modeled dissolved Fe(III) and Mn(II) concentrations."**

**Comments: 7.** With all the pollution in China, none of the models predict high pollution in Beijing which is surprising. How could it be explained?

**Response: This paper aims to investigate the potential role of different heterogeneous sulfate production mechanisms to explain this discrepancy. We have added to the introduction (P2L55) "Previous simulations have shown that most models fail to predict severe haze pollution in Beijing at least in part because of sulfate underestimation (Jiang et al., 2013; Park et al., 2014; Pozzer et al., 2012)."**

**Comments: 8.** In Figure 1, why the SOR is not correlated to SO4 during the 11/5 ?

**Response: Thanks for pointing this out. Beijing held the Asia-Pacific Economic Cooperation (APEC) Summit on 11/5-11, 2014. Stringent emission control measures were applied in Beijing and its surrounding regions to improve air quality (Zhang et al., 2016). We now state in the text P4L105 "During and before APEC, SO₂ emissions in Beijing and its surrounding regions decrease due to strict emission controls to improve air quality (Zhang et al., 2016; Liu et al., 2015b)." The increase in SOR (to ~30%) and stable sulfate concentrations during APEC likely suggest more efficient SO₂ oxidation.**